

# Measuring the spatiotemporal variability of snow depth in subarctic environments using unmanned aircraft systems (UAS) – Part 1: Measurements, processing, and accuracy assessment

Anssi Rauhala[1], Leo-Juhani Meriö[2], Anton Kuzmin[3], Pasi Korpelainen[3], Pertti Ala-aho[2], Timo Kumpula[3], Bjørn Kløve[2], Hannu Marttila[2]

[1]Civil Engineering, Faculty of Technology, University of Oulu, Oulu, FI-90570, Finland
[2]Water, Energy and Environmental Engineering, Faculty of Technology, University of Oulu, Oulu, FI-90570, Finland
[3]Department of Geographical and Historical Studies, University of Eastern Finland, Joensuu, FI-80101, Finland.

*Correspondence to*: Anssi Rauhala (anssi.rauhala@oulu.fi)

**Abstract.** Snow conditions in the northern hemisphere are rapidly changing, and information on snow depth is critical for decision-making and other societal needs. Unmanned aircraft systems (UASs) can offer data resolutions of a few centimeters at a catchment-scale, and thus provide a low-cost solution to bridge the gap between sparse manual probing and low-resolution satellite data. In this study, we present a series of snow depth measurements using different UAS platforms throughout the winter in the Finnish subarctic site Pallas, which has a heterogeneous landscape. We discuss the different platforms, the methods utilized, difficulties working in the harsh northern environment, and the results and their accuracy compared to in situ measurements. Generally, all UASs produced spatially representative estimates of snow depth in open areas after reliable georeferencing by using the Structure from Motion (SfM) photogrammetry technique. However, significant differences were observed in the accuracies produced by the different UASs compared to manual snow depth measurements, with overall RMSEs varying between 13.0 to 25.2 cm, depending on the UAS. Additionally, a reduction in accuracy was observed when moving from an open mire area to forest covered areas. We demonstrate the potential of low-cost UASs to efficiently map snow surface conditions, and we give some recommendations on UAS platform selection and operation in a harsh subarctic environment with variable canopy cover.

## 1 Introduction

Knowledge of changes in snow accumulation, depth and melt is crucial for nature and society in northern regions. In the northern hemisphere especially, snow is important to local ecology, communities, tourism and industry, supporting a unique environment in north and mountainous areas (Demiroglu et al., 2019; Boelman et al., 2019). Currently, snow resources are threatened by global warming, which will have many direct and indirect effects on northern environments (Carey et al., 2010, Bring et al., 2016). Any changes in magnitude, timing and variability of snowfall, accumulation patterns and melting will alter, among other things, water availability and soil moisture (Barnett et al., 2005; Kellomäki et al., 2010), which, in turn, impacts



flood prediction and warning, hydropower generation (reservoir inflow forecasting), water management, transportation, local
      authority daily management activities and the tourism sector (Veijalainen et al., 2010).

      Currently, snow depths are routinely monitored using snowlines (Lundberg et al., 2010, Stuefer et al., 2020), single or a
      network of multiple automatic stations (Zhang et al., 2017), or coarse satellite images (Frei et al., 2012). However, multiple
      studies have highlighted how measurements based on a few sampling locations do not provide a representative picture of the

wider spatial snow distribution (Grünewald and Lehning, 2015) and thus we lack reliable measures for spatially representative
      high-resolution snow depth information. Though technologies for manual sampling of snow depth exist (Sturm and Homgren,
      2018), on a wider scale manual measurements quickly become time-consuming and expensive. Modern remote sensing
      techniques can provide a cost-effective option for more spatially and temporally comprehensive snow depth measurements.

      The remote sensing of snow depth has traditionally utilized satellites or manned aircraft (Nolin, 2010; Dietz et al., 2012).

Satellite-derived snow depth products can offer global coverage, but the spatial resolution is low (Frei et al., 2012). Manned
      aircraft (e.g., airborne laser scanning, ALS), on the other hand, can provide better resolution with regional coverage (Deems
      et al., 2013, Currier et al., 2019), but the costs can be comparably high. Recently popularized unmanned aircraft systems
      (UASs) can offer resolutions of a few centimeters at a catchment-scale and thus provide a low-cost solution to bridge the gap
      between sparse manual probing and low-resolution satellite data.

Numerous studies have assessed the potential of using UASs in snow depth mapping in recent years (Vander Jagt et al., 2015;
      Bühler et al., 2016; De Michele et al., 2016; Harder et al., 2016; Lendzioch et al. 2016; Bühler et al., 2017; Cimoli et al., 2017;
      Adams et al., 2018; Avanzi et al., 2018; Fernandes et al., 2018; Redpath et al., 2018; Broxton et al. 2020; Harder et al. 2020;
      Revuelto et al., 2021). All the mentioned studies utilize structure from motion (SfM) photogrammetry, a low-cost survey
      technique developed from machine vision and traditional photogrammetry techniques, which has become widely popular for

geoscience applications such as topographic mapping (Westoby et al., 2012). The general approach when utilizing SfM
      photogrammetry in snow depth mapping is to produce at least two (snow-free and snow-covered) digital surface models
      (DSMs), then differentiate between the acquired models to estimate the snow depth.

      The majority of studies utilizing the UAS-SfM approach use a single UAS and either present results between only two surveys
      or are in relatively open areas in alpine, prairie and/or arctic settings, with only a few exceptions providing results for forested

areas (e.g., Lendzioch et al. 2016; Broxton et al. 2020) or providing a comparison between different UASs (Revuelto et al.
      2021). In this study, we present a series of snow depth measurements with different UAS platforms throughout the winter in
      the Finnish subarctic. We discuss the different platforms, the utilized methods, the difficulties of working in a harsh northern
      environment, and the results and their accuracy compared to in situ measurements. The accompanying paper (Meriö et al.
      2022, submitted to the same journal) delves deeper into the implications the data gathered has on local snow accumulation and

melting patterns. To our knowledge, we are providing the first series of snow depth measurements which rely on the UAS-
      SfM approach in a heterogeneous, subarctic, boreal, forest landscape and give a comparison of multiple UASs in variable
      lighting conditions and landscapes.



## 2 Data and methods

### 2.1 Study area

The study site (68.00° N, 24.21° E) is near the Pallas-Yllästunturi National Park in northern Finland, some 160 km north of the Arctic circle (Fig. 1A). The site consists of mostly mountaintop tundra, lower-elevation forests, wetlands, streams, and lakes (Aurela et al., 2015). The study site is part of the Pallas research catchment, which hosts multiple hydrological and meteorological observation stations (Marttila et al., 2021). The measurements were done within the Pallaslompolo catchment (total area 4.87 km², Fig. 1B), east of the Lommoltunturi and Sammaltunturi fells. The catchment ranges in altitude from 268

m to 375 m a.s.l and drains into a large lake, Pallasjärvi. The climate is characterized as a subarctic climate with persistent snow cover during the winter. The long-term (1981–2010) mean annual temperature and precipitation in the area are -1.0 °C and 521 mm, respectively (Pirinen et al., 2012). One-third of the annual precipitation falls as snow and the annual maximum snow depth, measured at the end of March, averages at 104 cm, while the mean annual snow water equivalent (SWE) was 200 mm in the period 1967-2020 (Marttila et al., 2021). Snow melting usually occurs during the second half of May or the

beginning of June and the first permanent snow typically appears in mid-October. Three subplots were chosen within the study site catchment to reduce the aerial mapping area to a more manageable size (Fig. 1B). Each subplot represents different land cover types in the area, one is a mostly open mire area (approx. 14.4 ha), one is a mostly coniferous forest-covered area (approx. 15.9 ha), and one is a mix between the two (approx. 15.4 ha).

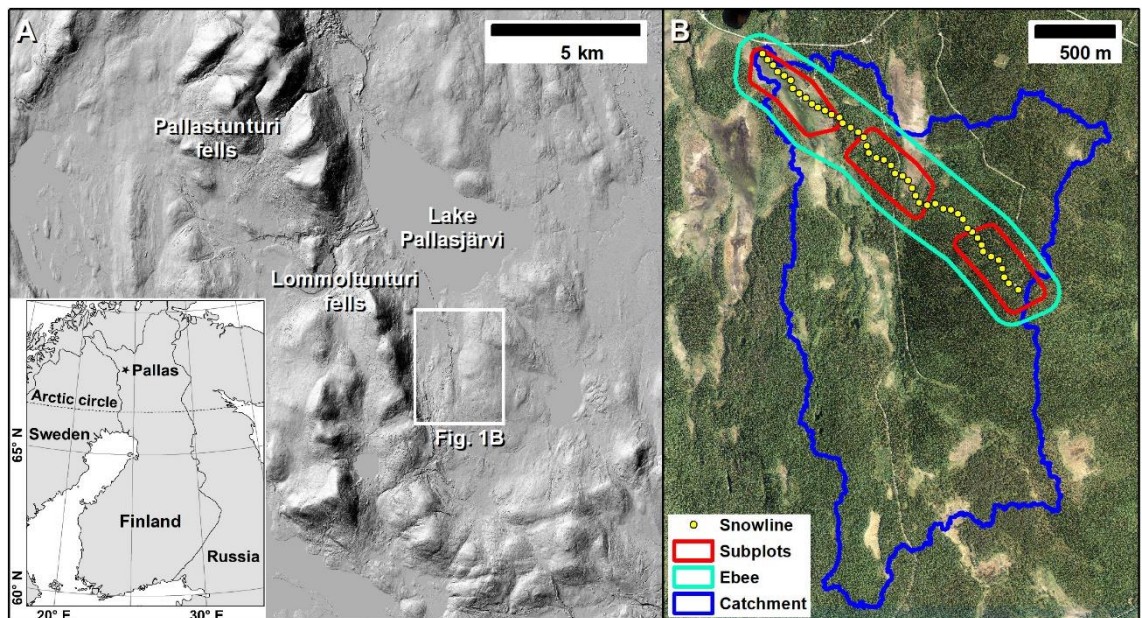


**Figure 1: A) Location of the study site south of Lake Pallasjärvi and east of the Lommoltunturi and Sammaltunturi fells. The location of Figure 1B is highlighted by the white rectangle. (Hillshade courtesy of National Land Survey of Finland.) B) Locations of the manual snowline measurements and outlines of the subplots (mire, mixed and forest read from northwest to southeast) and the eBee mapping area within the catchment. (Orthophoto courtesy of National Land Survey of Finland.)**



## 2.2 Equipment

Three quadcopters were utilized in mapping the subplots: a DJI Mavic Pro, a DJI Phantom 4 Advanced, and a DJI Phantom 4 RTK (Table 1). The Phantom 4 RTK represents the recently popularized UAS type which utilizes two GNSS (global navigation satellite system) receivers, one operating as a base station and one as a rover. Using a RTK (real-time kinematic) or PPK (post-processing kinematic) solution, positioning accuracy can achieve a level of a few centimeters, compared to the accuracy of a few meters obtained by the autonomously operating single-frequency GNSS receiver found in consumer-grade UASs such as the DJI Mavic Pro or the DJI Phantom 4 Advanced (e.g., Tomaštík et al., 2019). Besides the quadcopters, a fixed-wing SenseFly eBee Plus RTK drone was utilized during the four field campaigns to collect larger datasets encompassing either the subplots in a single flight or the whole catchment area (Fig. 1B).

**Table 1. The general specifications of the utilized UASs.**

| UAS | Mavic Pro | Phantom 4 Adv. | Phantom 4 RTK | eBee Plus RTK |
|---|---|---|---|---|
| **General portability** | High | Average | Average | Low |
| **Flight time (min)** | 27 | 30 | 30 | 59 |
| **Max. speed (km/h)** | 65 | 72 | 72 | 110 |
| **Camera CMOS sensor size** | 1/2.3" | 1" | 1" | 1" |
| **Effective megapixels** | 12.3 | 20 | 20 | 20 |
| **Focal length (mm, 35 mm equiv.)** | 26 | 24 | 24 | 29 |
| **Field of view (°)** | 79 | 84 | 84 | 64 |
| **Ground resolution at 100 m (cm/pixel)** | 3.1 | 2.7 | 2.7 | 2.3 |

The DJI Mavic Pro quadcopter is a comparably small UAS with a weight of less than 800 g and a folded size of 8.3 x 8.3 x 19.8 cm; it thus represents a very portable option considering aerial mapping in areas that are difficult to traverse, especially during the winter months. The Mavic Pro has a comparably small 1/2.3" sensor with 12.3 megapixels and a 26 mm (35 mm equivalent) lens, with the focus being mostly on portability.

The DJI Phantom 4 Advanced and the DJI Phantom 4 RTK quadcopters are roughly 19 x 29 x 29 cm in size and weigh around 1.4 kg, and both have 1" sensors with 20 megapixels and a 24 mm (35 mm equivalent) lens. The DJI Phantom 4 quadcopters represent average sized UASs when considering the portability in this context. Compared to the Mavic Pro, the Phantom 4 quadcopters have an increased maximum speed (65 vs. 72 km h$^{-1}$), increased maximum flight time (27 vs. 30 min), and arguably better wind resistance. The larger sensor size of the Phantoms also improves their light gathering ability, which is generally linked to improved image quality.

The SenseFly eBee Plus RTK fixed-wing has a wingspan of 110 cm and a weight (incl. camera) of around 1.1 kg. The large size greatly reduces the portability of the eBee. However, being a fixed-wing UAS, the eBee has a greatly improved flight time





(59 min) and maximum speed (110 km h$^{-1}$), and consequently has a greatly improved areal coverage when compared to the

quadcopters. The eBee was equipped with a SenseFly S.O.D.A. with a 1" 20-megapixel sensor and a 29 mm (35 mm
equivalent) lens. As the eBee Plus RTK fixed-wing cannot ascend and descend vertically, it requires a comparably flat and
large clearing for takeoff and landing, when compared to the quadcopters.

External ground control points (GCPs) and an RTK GNSS receiver to measure location are needed for rectifying the gathered
aerial imagery, especially with UASs which are not equipped with internal RTK correction. In this study, a Trimble R10 and

a Topcon Hiper-V RTK GNSS receiver were utilized for measuring the GCP locations. The necessity of marking and
measuring the GCPs can be burdensome for the field crew, especially during the polar nights, when the time window for flights
is short and the GCPs might have to be crafted in the dark. Furthermore, carrying the extra equipment severely reduces the
portability of the unmanned aerial system (UAS), although, with a field crew of two or more people, it is somewhat easier to
divide the load or use a sled, even when equipped with skis. The RTK-equipped UASs also require an RTK base station.

However, as static equipment which can be placed into a suitable location, the RTK system is much less work-intensive,
although the purchasing cost of it is many times higher.

**2.3 UAS data acquisition and field data collection**

Eight UAS campaigns were carried out at the site between June 2018 and June 2019 (Table 2). The utilized equipment varied
between different campaigns, depending on the availability of UASs and field crews. Of the eight campaigns, five, from

December 2018 to April 2019, were during the snowy season. During the January campaign, only the mire subplot was mapped
with the DJI Mavic Pro, and the data was of poor quality due to the camera lens mechanism freezing in very low temperatures,
which reached -30 °C. During the April 22–25 campaign, the data from eBee Plus RTK was lost due to a likely electronic
malfunction. The target ground sampling distances (GSD) for the UASs were 3.7 cm for the DJI Mavic Pro, 3.0 cm for the
DJI Phantom 4 Advanced, 3.0 cm for the DJI Phantom 4 RTK, and 4.5 cm for the eBee Plus RTK. Slight variations in the

target GSDs were inherently required due to different sensors/focal lengths and different flight heights because the aim was to
demonstrate typical use case scenarios, e.g., ability to capture a subplot (or all subplots in the case of the eBee) with a single
battery. The targets for forward and side overlap were a minimum of 80 % and 75 %, respectively.

With the non-RTK UAS, GCPs were utilized to rectify the models. The number of GCPs varied slightly for different sites and
campaigns, as the low light conditions during the polar night and the peak snow depth later in the spring limited the mobility

of the field crew. During the snow-free surveys, painted plywood targets were utilized as GCPs, and during the snow-covered
surveys, the targets were painted straight onto the snow surface. In addition to the temporary GCPs, six permanent GCPs were
installed on top of ~2 m wooden posts and were tested in each subplot. On average, 13 temporary GCPs (8–17, median 13.5)
and 6 permanent GCPs (PGCPs) were utilized to rectify the non-RTK data (Fig. 3) during the winter campaigns. The summer
campaign of June 2018 utilized 18–21 GCPs per subplot. For the RTK UAS, a single GCP was used for correcting elevation

bias.  Also, an average of 16 elevation checkpoints (6–38, median 14.5) were measured at random locations every time at each
plot to get a rough estimate of the external accuracy of the generated DSMs compared to the RTK GNSS reference elevation.



Besides the gathered UAS data, we also utilized airborne laser scanning (ALS) data with a reported maximum vertical standard error of 15 cm, collected by the National Land Survey of Finland during the summer of 2018 as part of the national survey campaign. The DEM based on ALS data has a ground resolution of 1 m/pixel. Snow depth reference data was collected from

a snow course passing through all the subplots, consisting of 46 fixed snow stake measurement points, placed an average of 52 meters apart (standard deviation 6.2 m), of which 35 were within the subplots (Fig. 1B). Snow depth reference data was also available through the use of an automatic ultrasonic snow depth sensor (Campbell Scientific SR50-45H) with an accuracy of ±1 cm, located in the forest subplot at the highest elevation of the study area and operated by the Finnish Meteorological Institute.


**Table 2. UASs utilized during different campaigns.**

| UAS campaign | DJI Mavic Pro | DJI Phantom 4 Adv. | DJI Phantom 4 RTK | eBee Plus RTK |
|---|---|---|---|---|
| 12.-15.06.2018 | x | | | |
| 10.-13.12.2018 | x | x | x | x |
| 21.-25.01.2019 | x* | | | |
| 18.-22.02.2019 | x | | x | x |
| 01.-05.04.2019 | x | | x | x |
| 22.-25.04.2019 | x | | x | x** |
| 20.-21.05.2019 | x | x | | |
| 04.-05.06.2019 | | | x | x |

*) data was of poor quality, **) data was lost

**2.4 Data processing and analysis**

The acquired aerial data was processed using Agisoft Photoscan/Metashape Professional v1.4.5./v.1.6., which employs the
SfM technique to produce orthomosaics and DSMs. The SfM technique is described in detail by Westoby et al. (2012). To better harmonize the data, each dataset was processed using high quality and moderate depth filtering settings. The produced orthomosaics and DSMs were further processed in ESRI ArcGIS 10.6 (Fig. 2). Due to the poor sub-canopy penetration of the SfM technique (Harder et al., 2020), masking was used to omit data at the locations and in the immediate vicinity of trees (Fig. 3). The three masks were generated using Maximum Likelihood Supervised Classification. The supervised classification was
based on the orthomosaics from the 3 April 2019 survey that had snow-free tree canopies, thus showing the high contrast between the canopies and snowy ground. Further analysis revealed that sometimes the SfM technique struggled with depth mapping of deciduous and snow-covered trees, thus leading to artificially high snow depths immediately next to trees/masks. To mitigate the errors, the masks were further buffered by 36 cm, which was found to be a good compromise for removing artificially high values without losing too much data close to the trees.






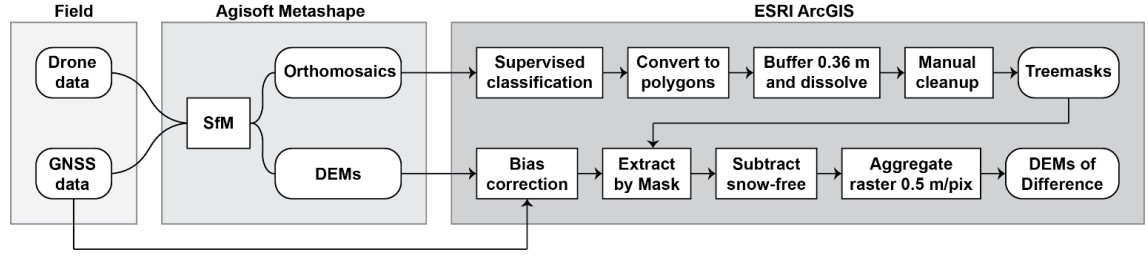

**Figure 2: Flowchart for the handling and analysis of the UAS datasets.**

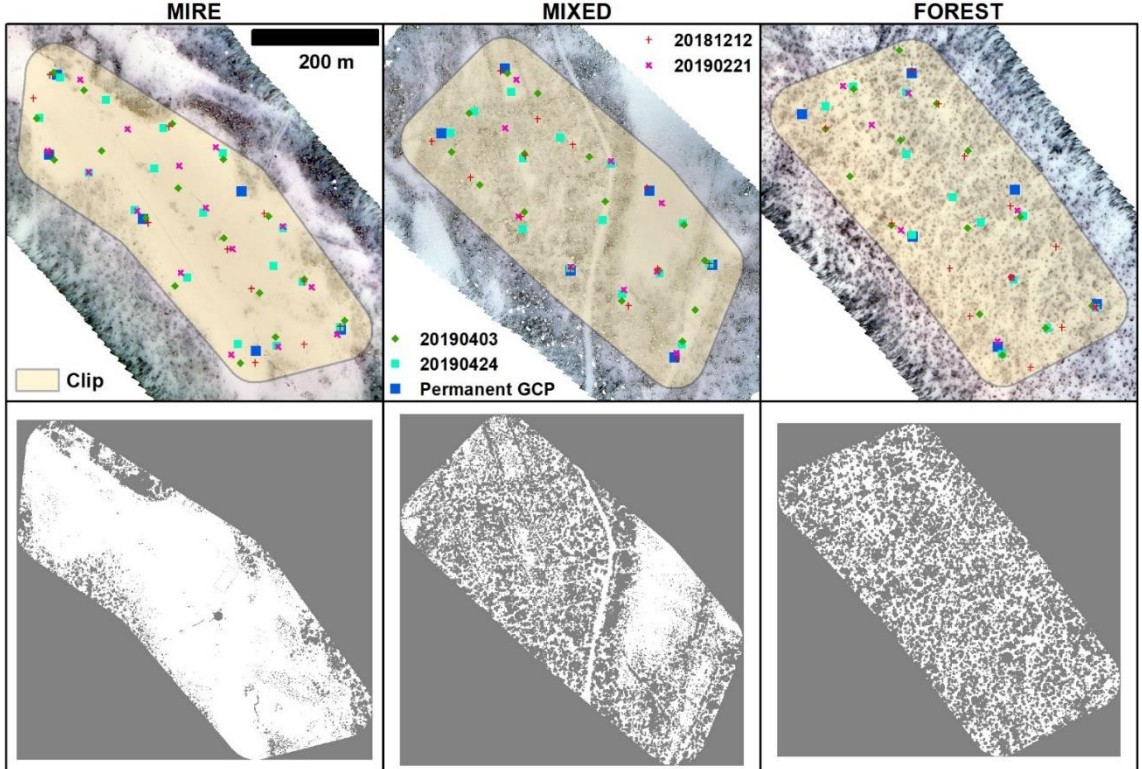

**Figure 3:** *Above:* **The subplots and locations of the permanent and temporary ground control points utilized during the successful winter measurement campaigns.** *Below:* **Tree masks generated for each subplot.**

After manual cleanup of a few classification errors, the masks were utilized for canopy removal before subtracting the snow-free (bare ground) DSM from each snow-covered DSM. Finally, the DEMs of difference (DoD) highlighting the snow depth
were aggregated to 50 cm per pixel resolution before further analysis. The aggregation allows us to smoothen some small-scale variability while retaining a reasonable resolution for the snow-vegetation interaction analysis discussed in the accompanying paper (Meriö et al. 2022, submitted). The 50 cm per pixel resolution was decided as a good middle ground



following the findings of De Michele et al. (2016), who demonstrated how the standard deviation of UAS-derived snow depth increases with increasing resolution but stabilizes at a $\leq 1$ m pixel size.

To estimate the uncertainty of generated DSMs, the difference between UAS and RTK GNSS elevation $\Delta z$ at each checkpoint was calculated following Eq. (1):

$$\Delta z_t = DSM_{S,t} - z_{CP,t} \,, \tag{1}$$

where t is the date of survey, $DSM_S$ is the snow surface elevation from the UAS survey, and $z_{CP}$ is the checkpoint elevation measured with RTK GNSS. Considering error propagation when differentiating between two DSMs (e.g., Brasington et al.,

2003), the precision of the DoDs highlighting the snow depth was estimated following Eq. (2):

$$u = \sqrt{\sigma(\Delta z_t)^2 + \sigma(\Delta z_G)^2} \,, \tag{2}$$

where $\sigma(\Delta z_t)$ is the standard deviation for the difference between UAS and RTK GNSS elevation $\Delta z$ for each winter survey, and $\sigma(\Delta z_G)$ is the standard deviation for the difference between UAS and RTK GNSS elevations for the snow free ground DSM. Since the DSMs are not free of bias (i.e., mean error), error propagation for mean errors highlighting the trueness of

DoDs were calculated following Eq. (3):

$$m = \mu(\Delta z_t) + \mu(\Delta z_t) \,, \tag{3}$$

where $\mu(\Delta z_t)$ is the mean error for the difference between UAS and RTK GNSS elevation $\Delta z$ for each winter survey and $\mu(\Delta z_G)$ is the mean error for the difference between UAS and RTK GNSS elevations for the snow-free ground DSM. Snow depth for each pixel $hs_{DSM}$ was calculated following Eq. (4):

$$hs_{DSM,t} = DSM_{S,t} - DSM_G \,, \tag{4}$$

where $DSM_S$ is snow surface elevation from the UAS survey, and $DSM_G$ is the snow-free ground elevation from UAS/ALS survey. The difference between UAS-derived snow depth and manual snowline measurements $\Delta hs$ was calculated following Eq. (5):

$$\Delta hs_t = hs_{DSM,t} - hs_{SL,t} \,, \tag{5}$$

where $hs_{SL,t}$ is the manual snow depth measurement.

**3 Results**

Figures 4 and 5 show the resulting orthomosaics and snow depth maps for different subplots generated from the data collected with the DJI Phantom 4 RTK on the dates of 12 December 2018 (DEC 12), 21 February 2019 (FEB 21), 3 April 2019 (APR 03), and 24 April 2019 (APR 24). Similar maps were also produced for the other UAS data when available (see Table 2).






**Figure 4: Orthomosaics for different subplots produced from the DJI Phantom 4 RTK data obtained during the winter surveys.**







**Figure 5: Snow depth maps for different subplots produced from the DJI Phantom 4 RTK data obtained during the winter surveys.**





### 3.1 Comparison to GNSS checkpoints

Figure 6A combines measurements from the FEB 21 and APR 03 surveys using the DJI Phantom 4 RTK (P4RTK), DJI Mavic Pro (Mavic), and eBee Plus RTK (eBee) to highlight the effects of land cover on the difference between the DEM and GNSS survey elevations as calculated following Eq. (1). Separate boxplots for each subplot and date are provided in the supplementary material (see Fig. S1 in the Supplement), including results for DJI Phantom 4 Advanced (P4A), which was only utilized during the DEC 12 winter survey. There is a general trend of increase in the differences between the DEM and GNSS survey elevations when moving from the mire subplot to mixed and forest subplots, which is observable with all the UAS. Based on Levene's test, there are statistically significant differences in sample variances at a significance level of 0.05 with each UAS. When comparing the UASs to each other, there are statistically significant differences in the mire and forest subplots, where P4RTK and eBee produce fairly similar data, but Mavic is clearly the least accurate. In each case, P4RTK has the best error statistics.

Figure 6B combines all subplots for the DEC 12, FEB 21, and APR 03 surveys to highlight the effect of date (lighting conditions) on the uncertainty of UAS measurements compared to checkpoints. There are no statistically significant differences in sample variances with any UAS from the DEC 12 lowlight polar night conditions and the FEB 22 and APR 03 measurements, although there is a slight improvement (~1–2 cm) in mean absolute error (MAE) from DEC 12 to FEB 22 with each UAS. However, when the UASs are compared to each other, P4RTK produced the most accurate data in each case and statistically significant differences in variance can be observed in the DEC 12 and FEB 21 surveys, where P4RTK and eBee are again fairly similar, but Mavic struggles.





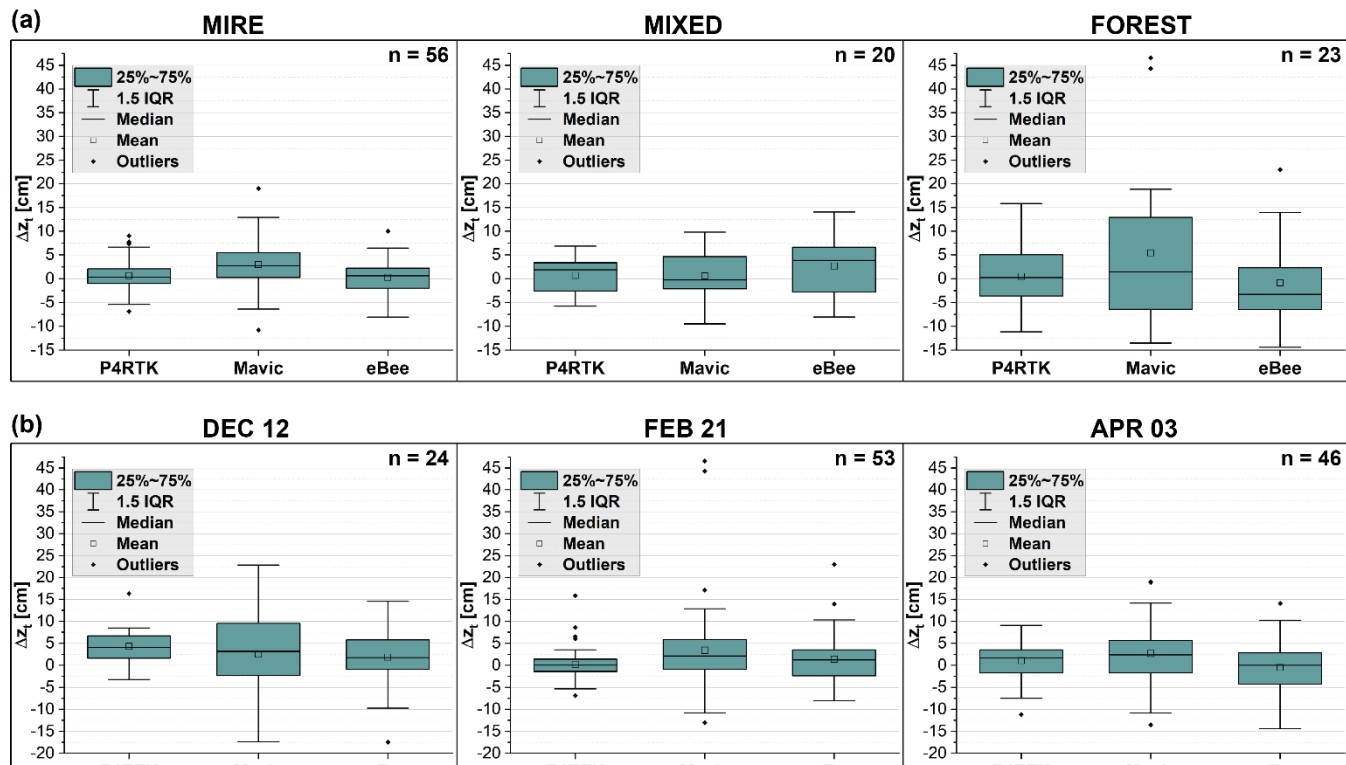

**Figure 6. A) Difference between the DEM and GNSS survey elevations for each subplot from the FEB 21 and APR 03 surveys. B) Difference between the DEM and GNSS survey elevations for the DEC 12, FEB 21 and APR 03 surveys from all subplots.**

Table 3 highlights the precision of snow depth DoDs calculated following Eq. (2). With P4RTK, the precision is most stable in the mire subplot (in the <4.5 cm range), whereas there is slightly more variation in the mixed and forest subplots, ranging from 3.98 cm to 7.18 cm. With Mavic, the precision is in each case lower than with P4RTK or eBee, and there is a tendency for accuracies to decrease from the mire subplot to the mixed and forest subplots. With eBee, the precision is very similar to P4RTK in the mire subplot. However, there is also a general tendency for precision to decrease towards mixed and forest subplots with eBee, the exception being the APR 03 dataset, during which eBee also provides slightly better precision in the mire subplot when compared to P4RTK. Table 4 highlights the trueness of snow depth DoDs calculated following Eq. (3). Again, P4RTK provides the best accuracy overall, with the only exception being the DEC 12 survey in the forest subplot. With eBee, the trueness suffers from large biases of ~16.6 cm and ~10.0 cm for the snow-free bareground DSMs for the mire and mixed subplots, respectively.



**Table 3. DoD precision during different measurement periods and in study locations.**

|  | MIRE | | | MIXED | | | FOREST | | |
|---|---|---|---|---|---|---|---|---|---|
|  | **P4RTK** | **Mavic** | **eBee** | **P4RTK** | **Mavic** | **eBee** | **P4RTK** | **Mavic** | **eBee** |
| **DEC12 (cm)** | 4.18 | 7.76 | 4.53 | 6.54 | 13.19 | 7.87 | 4.16 | 11.59 | 9.20 |
| **FEB21 (cm)** | 4.13 | 6.93 | 5.16 | 4.70 | 10.67 | 6.17 | 7.18 | 21.87 | 9.40 |
| **APR03 (cm)** | 4.48 | 7.17 | 4.13 | 5.71 | 9.67 | 8.66 | 5.34 | 10.23 | 5.66 |
| **APR24 (cm)** | 4.08 | 6.04 |  | 4.98 | 10.26 |  | 3.98 | 8.05 |  |

**Table 4. DoD trueness during different measurement periods and in study locations.**

|  | MIRE | | | MIXED | | | FOREST | | |
|---|---|---|---|---|---|---|---|---|---|
|  | **P4RTK** | **Mavic** | **eBee** | **P4RTK** | **Mavic** | **eBee** | **P4RTK** | **Mavic** | **eBee** |
| **DEC12 (cm)** | 2.85 | 11.91 | 19.94 | 7.24 | 8.11 | 12.23 | 7.25 | 3.78 | 4.56 |
| **FEB21 (cm)** | -1.32 | 10.07 | 16.60 | 3.26 | 5.04 | 11.69 | 7.18 | 14.96 | 11.77 |
| **APR03 (cm)** | 2.86 | 12.36 | 17.34 | 2.10 | 6.49 | 13.13 | 2.56 | 4.45 | -0.92 |
| **APR24 (cm)** | -0.70 | 8.16 |  | 3.03 | 9.48 |  | 1.28 | 9.20 |  |

## 3.2 Comparison to manual snowline measurements

Manual snowline measurements resulted in mean snow depths and standard deviations of 36.8 cm and 4.8 cm for the DEC 12 survey, 76.5 cm and 4.9 cm for the FEB 21 survey, 86.9 cm and 9.1 cm for the APR 03 survey, and 35.8 cm and 20.6 cm for the APR 24 survey. A general trend of increasing snow depth variation in the landscape was observed as winter progressed, indicated by the standard deviations. During the APR 24 survey, the variation was high due to the spring melt and resulting flooding already being especially pronounced in the mire area. If the mire area is ignored, the mean snow depth and standard deviations were 46.2 cm and 10.5 cm for APR 24. Figure 7 shows examples of snow depth distributions, along with manual snowline and single automatic ultrasonic snow depth measurements during different surveys, obtained using the P4RTK data. The histogram shapes are generally long-tailed normal distributions. The biggest deviances from a normal distribution are seen on the mire and mixed subplots during the APR 24 spring melt.






**Figure 7. Snow depth histograms for different subplots based on P4RTK data during the DEC 12, FEB 21, APR 03, and APR 24 surveys. Vertical lines indicate the median snow depth for each subplot. The boxplots indicate the results from manual snowline measurements and the red dot indicates data from an automatic ultrasonic snow depth sensor located in the Forest subplot.**





Table 5 shows the statistics for differences between the UAS-derived snow depths and the manual snow line measurements calculated following Eq. (4). The data is provided for snow depth DoDs utilizing UAS-derived snow-free DEM and ALS-derived snow-free DEM. It should be noted that the values do not account for any potential systematic or random errors in the manual snowline measurements. The snow depth measurements with P4RTK are of practically equal accuracy regardless of whether UAS or ALS data is used as the snow-free bare ground model. With Mavic and eBee, utilizing the ALS bare ground

model produced generally more accurate results. Mavic and eBee tend to produce a higher number of outlier magnitude errors when compared to P4RTK, which clearly generates the most accurate data of the three, especially when UAS data is used as a snow-free model.

**Table 5. The difference between UAS-derived snow depths and manual snow line measurements for the combined winter dataset.**
**Mean errors (ME), mean absolute errors (MAE), standard deviations (SD), root mean square errors (RMSE), minimum errors (MIN), and maximum errors (MAX) are provided for snow depth DoDs utilizing UAS-derived snow-free DEM and ALS-derived snow-free DEM (indicated by the -L postfix).**

|  | P4RTK | P4RTK-L | Mavic | Mavic-L | eBee | eBee-L |
|---|---|---|---|---|---|---|
| **n** | 140 | 140 | 140 | 140 | 105 | 105 |
| **ME (cm)** | 3.9 | 3.9 | -1.4 | 7.4 | -5.7 | 3.4 |
| **MAE (cm)** | 9.7 | 10.3 | 18.4 | 13.2 | 16.4 | 10.7 |
| **SD (cm)** | 12.4 | 13.1 | 25.1 | 19.2 | 22.6 | 13.6 |
| **RMSE (cm)** | 13.0 | 13.7 | 25.2 | 20.6 | 23.3 | 14.1 |
| **MIN (cm)** | -18.7 | -22.3 | -49.6 | -24.0 | -94.9 | -30.0 |
| **MAX (cm)** | 54.8 | 57.0 | 104.0 | 102.0 | 43.0 | 52.7 |

Not including outliers, the greatest differences between the snowline and UAS-derived snow depths were observed in the mire
subplot during the APR 24 survey. These differences are most probably related to the spring melt and flooding that was pronounced on the mire subplot (see Fig. 4), which resulted in some of the snowline points being under a mixture of muddy water and ice. In the manual snowline survey, these points were marked as having zero snow depth. However, the UAS-approach, based on SfM photogrammetry and differentiating between the DEMs, produces average snow depths of 23–35.6 cm for these points. Table 6 shows the statistics for differences between the UAS-derived snow depths and manual snow line

measurements when the APR 24 survey data is removed from the dataset, i.e., when all UASs have an equal amount of data points. The removal of the APR 24 data, including the spring melt data points from the mire subplot, further demonstrates the better accuracy of the P4RTK compared to the other UASs.





**Table 6. Difference between UAS-derived snow depths and manual snow line measurements for the DEC 12 - APR 03 dataset. Mean errors (ME), mean absolute errors (MAE), standard deviations (SD), root mean square errors (RMSE), minimum errors (MIN), and maximum errors (MAX) are provided for snow depth DoDs utilizing UAS-derived snow-free DEM and ALS-derived snow-free DEM (indicated by the -L postfix).**

|  | P4RTK | P4RTK-L | Mavic | Mavic-L | eBee | eBee-L |
|---|---|---|---|---|---|---|
| **n** | 105 | 105 | 105 | 105 | 105 | 105 |
| **ME (cm)** | 2.1 | 2.1 | -2.7 | 6.0 | -5.7 | 3.4 |
| **MAE (cm)** | 8.7 | 8.9 | 18.8 | 12.6 | 16.4 | 10.7 |
| **SD (cm)** | 10.5 | 10.7 | 26.1 | 19.6 | 22.6 | 13.6 |
| **RMSE (cm)** | 10.7 | 10.9 | 26.3 | 20.4 | 23.3 | 14.1 |
| **MIN (cm)** | -18.7 | -22.3 | -49.6 | -24.0 | -94.9 | -30.0 |
| **MAX (cm)** | 33.7 | 32.0 | 104.0 | 102.0 | 43.0 | 52.7 |

No clear correlation was observable on individual subplots/dates when comparing manual snow depth measurements and the UAS-derived snow depth pixels at the location of the manual measurement. However, when all subplots from different field trips were combined, significant Pearson correlations were observed for each UAS at a significance level of 0.05. The correlations between UAS and field measurements were 0.89 for P4RTK, 0.64 for Mavic, and 0.60 for eBee, when utilizing UAS data as a snow-free bare ground model. Correspondingly, the correlations were 0.87 for P4RTK, 0.74 for Mavic, and 0.81 for eBee when utilizing ALS data for the snow-free model.

Figure 8A combines measurements from the DEC 12, FEB 21, and APR 03 surveys to highlight the effect of land cover, and Fig. 8B combines all subplots to highlight the effect of survey date on the difference between UAS-derived snow depths and manual snow line measurements when UAS data is used as the snow-free bare ground model. Figures 9A and 9B provide corresponding data for when ALS data is used as the snow-free model. Separate boxplots for each subplot and date are provided in the supplementary material (Fig. S3).

When utilizing UAS data as a snow-free model, there are statistically significant differences in sample variances with respect to land cover (Fig. 8A) for each UAS, but there are no significant differences concerning survey date (Fig. 8B). A similar trend that was observed with the GNSS checkpoints can also be seen, with accuracy decreasing when moving from mire to mixed/forest subplot. When comparing the UAS to each other, there are statistically significant differences for each subplot and survey date with P4RTK always producing the most accurate data and eBee producing the second best, aside from the mixed subplot, where a large bias was observed with eBee when comparing the DEM/DoD data on GNSS checkpoints (Table 4).



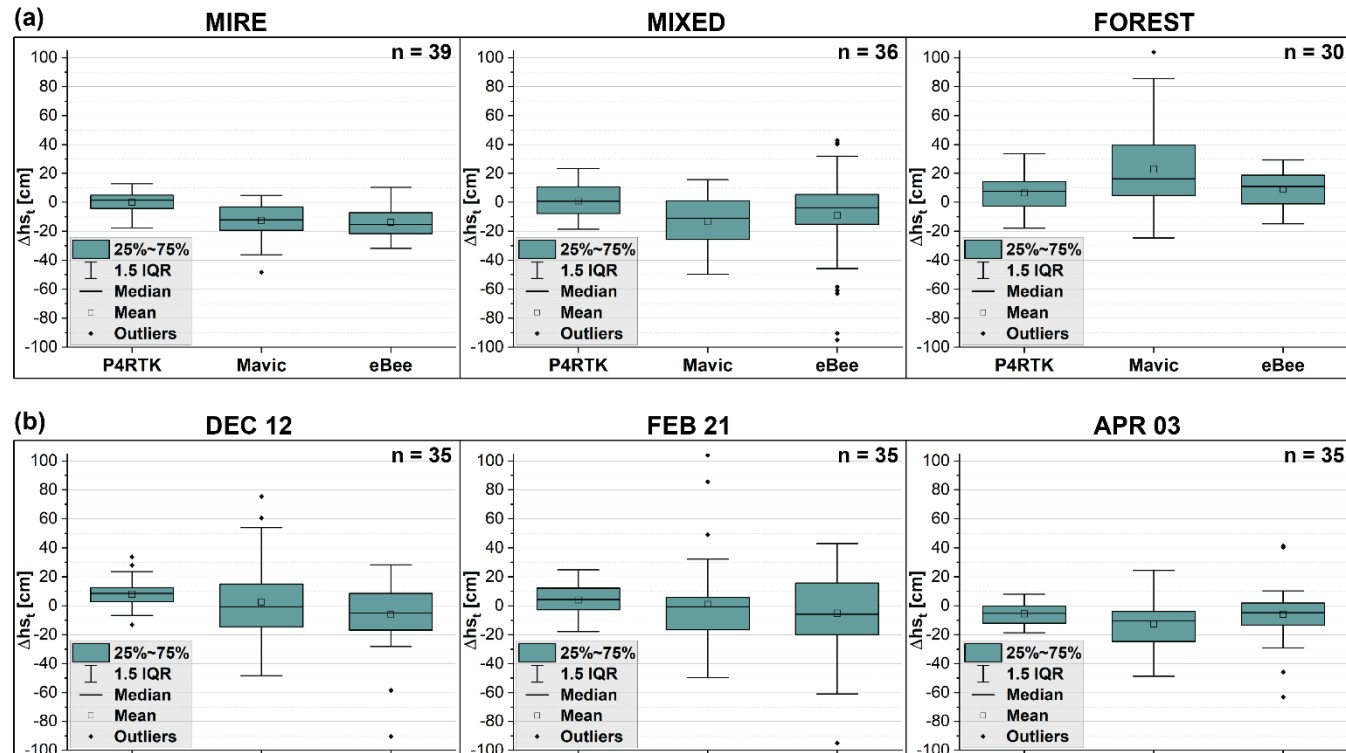

**Figure 8.** A) Difference between the snow depth DoDs and manual snow line measurements for each subplot from the DEC 12, FEB 21, and APR 03 surveys utilizing UAS-derived snow-free DEM. B) Difference between the snow depth DoDs and manual snow line measurements for the DEC 12, FEB 21 and APR 03 surveys from all subplots utilizing UAS-derived snow-free DSM.

When ALS data is used as a snow-free model, there are similarly significant differences in sample variances to land cover with each UAS (Fig. 9A), but also for survey date with Mavic and eBee (Fig. 9B). Again, the accuracies decrease with a move from mire to mixed/forest. With respect to survey date, Mavic performed poorly during the DEC 12 and FEB 21 surveys, and eBee performed poorly during the FEB 21 survey. When comparing the UASs to each other, statistically significant differences in variances between the UASs are observed only with the forest subplot and DEC 12 survey. In both cases, Mavic clearly produces the least accurate data. In general, the utilization of ALS data as a snow-free model clearly benefits Mavic and eBee in all situations, whereas there is no practical difference with P4RTK, which again produces the most accurate date in each case. However, with the ALS, the accuracies of eBee and Mavic are in general much closer to P4RTK. Table 7 shows the RMSEs for the data displayed in Fig. 8 and Fig. 9 for a comparison with existing literature discussed in the next section.





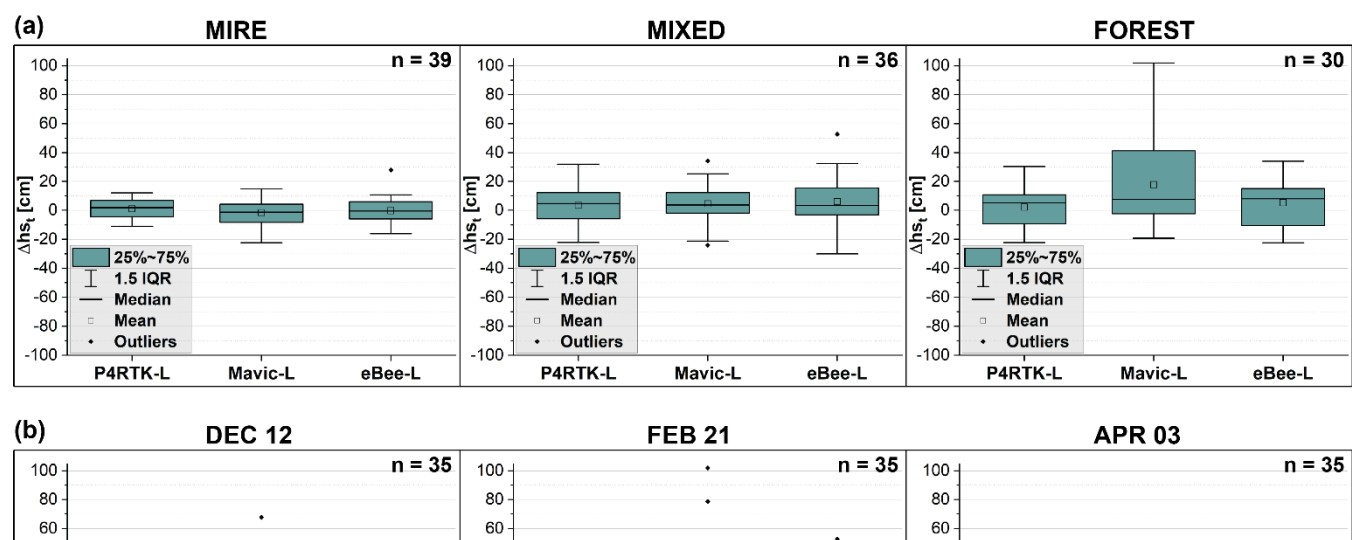

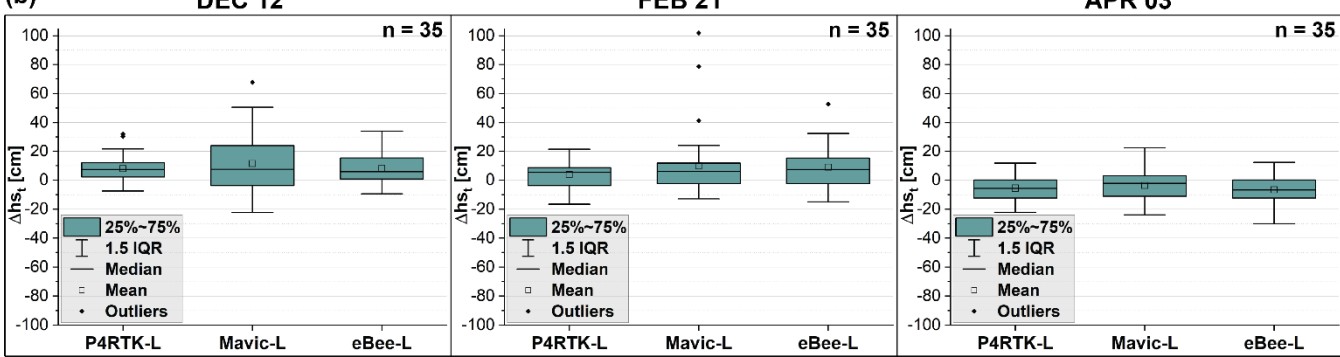

**Figure 9. A) Difference between the snow depth DoDs and manual snow line measurements for each subplot from the DEC 12, FEB 21, and APR 03 surveys utilizing ALS-derived snow-free DEM. B) Difference between the snow depth DoDs and manual snow line measurements for the DEC 12, FEB 21 and APR 03 surveys from all subplots utilizing ALS-derived snow-free DSM.**


**Table 7. Root mean square errors (RMSE) in cm for the difference between UAS-derived snow depths and manual snow line measurements for the different subplots (DEC 12 - APR 03 surveys combined) and different survey dates (subplots combined).**

|  | P4RTK | P4RTK-L | Mavic | Mavic-L | eBee | eBee-L |
|---|---|---|---|---|---|---|
| **Mire** | 7.2 | 6.6 | 17.2 | 8.4 | 17.4 | 8.6 |
| **Mixed** | 10.6 | 13.0 | 21.9 | 13.1 | 32.4 | 16.0 |
| **Forest** | 14.1 | 12.6 | 38.1 | 34.2 | 14.7 | 16.9 |
| **DEC 12** | 12.0 | 11.9 | 27.8 | 23.2 | 22.8 | 12.9 |
| **FEB 21** | 10.7 | 10.5 | 29.9 | 24.9 | 27.1 | 17.1 |
| **APR 03** | 9.1 | 10.3 | 20.8 | 10.0 | 19.5 | 11.5 |



## 4 Discussion

### 4.1 Accuracy of UAS based snow depth measurements for different platforms

When considering the SfM-approach and the resulting model accuracy to detect snow depth, the biggest differences in the specifications of the UAS and flight parameters used in the study were between the camera sensors and focal lengths, and the utilized flight height and georeferencing method. The flight heights had to be optimized to be able to fly individual missions in 15–20 minutes with the multicopters to account for the reduced battery performance in cold weather. This resulted in slight differences in the resulting ground sample distances between the UASs. However, by far the greatest differences were in the

georeferencing methods, with Mavic (and P4A) relying solely on GCPs, and P4RTK and eBee relying more on RTK correction to ensure high positional accuracy of UAS with a single GCP utilized to mitigate possible elevation bias.

The accuracy of GCP-based georeferencing and the suitable number of GCPs have been discussed by several authors (Tonkin and Midgley, 2016; Martínez-Carricando et al., 2018; Sanz-Ablanedo et al., 2018; Yu et al., 2020). Generally, such assessments are done in open area locations, where the planning and distribution of the GCP network is relatively easy. In complex, forested

environments with thick snow cover, the distribution of GCPs is generally more challenging, considering the lack of mobility, time constraints, and a lack of open areas with an unrestricted view of the sky. Nevertheless, these kinds of studies can be used as a baseline to assess the quality of GCP placement in this study. The importance of uniform GCP distribution for high data accuracy was highlighted by Tonkin and Midgley (2016), although they also note that excess GCPs lead to diminishing returns. Sanz-Ablanedo et al. (2018) demonstrated that a planimetric RMSE accuracy similar to ±GSD was achieved with approx. 2.5–

3 GCPs per 100 photos. Vertical accuracy improved towards 1.5 x GSD when using more GCPs, with the maximum in their tests being achieved with 4 GCPs per 100 photos. Martínez-Carricando et al. (2018) demonstrated that with stratified GCP distribution, there was a clear improvement in vertical accuracy when moving from 0.25 GCP to 1 GCP per ha (RMSE from 30.8 cm to 5.2 cm), whereas moving from 1 GCP to 2 GCP per ha (RMSE 4.3 cm) yielded diminishing returns. Yu et al. (2020) argued that for survey areas of 7–39 ha, a minimum of 6 GCPs was required for an accuracy of 10 cm and more than 12 GCPs

were required for optimal results.

In recent years, multiple studies have also compared DSM accuracies produced by GCP georeferencing and DSM accuracies produced through RTK/PPK solutions (Benassi et al., 2017; Forlani et al., 2018; Bolkas, 2019; Padró et al., 2019; Tomaštík et al., 2019; Zhang et al., 2019). Tomaštík et al. (2019) compared the PPK approach to 4 GCP and 9 GCP georeferencing under two different canopy conditions (study area approx. 270 ha) and concluded that the PPK approach offered better or equal

accuracy and was not influenced by vegetation seasonal variation, unlike GCP georeferencing. Zhang et al. (2019) compared PPK to 8 GCP georeferencing on cultivated land (study area approx. 1.7 ha) before and after plowing and concluded that a PPK solution produces the same accuracy as the GCP approach, but a single GCP is necessary to correct possible vertical bias. Comparing the RTK approach to 12 GCP georeferencing in an urban area (approx. 18 ha) with buildings, roads, car parks and meadows, Forlani et al. (2018) had similar results showing how RTK can offer a similar accuracy when at least one GCP is

utilized to correct vertical bias.





The three subplots in this study were between approx. 14.4–15.9 ha, and the average number of GCPs was 13 per subplot (8–17 depending on the survey), resulting in an average of 5.1 GCPs per 100 photos and 0.87 GCP per ha, or 7.38 GCPs per 100 photos and 1.27 GCP per ha when the six PGCPs are also included. Thus, the amount of GCPs utilized was within the range suggested by Sanz-Ablanedo et al. (2018), Martínez-Carricando et al. (2018), and Yu et al. (2020). Furthermore, there was no
significant Kendall's rank correlation between the number of GCPs utilized and the accuracy (e.g., standard deviation or mean absolute error) of snow depth measurements with Mavic, which relied solely on GCP georeferencing. Nevertheless, with regards to accuracy, P4RTK utilizing single GCP clearly outperformed Mavic utilizing all available GCPs and PGCPs, although it should be noted that Mavic has a smaller sensor size and had slightly lower (~0.7 cm) GSD. More surprisingly, the accuracy of P4A utilized during the DEC 12 surveys was clearly worse compared to P4RTK and was more in line with Mavic
data (see Supplements), although the two Phantoms have the same sensor size and focal length, and had practically the same GSD. This might indicate that RTK-supported data acquisition outperforms the traditional GCP-based method in these conditions even when only utilizing a single GCP with the RTK. This is somewhat in line with the results of Tomaštík et al. (2019), who reported that the PPK approach was not influenced by seasonal variation of vegetation, unlike the GCP georeferencing approach. However, it should also be noted that the DEC 12 measurements were made during lowlight polar
night conditions in which a comparably short time difference between the subsequent flights could significantly affect the amount of available light. The eBee data shows a clear negative bias in the mire subplot and on some occasions positive bias in the forest subplot (see Fig. S2 in the Supplement), possibly indicating small orientational errors in the datasets acquired for the whole catchment. Thus, for large datasets, a single GCP might not be sufficient even with RTK-equipped UAS.

Processing the non-RTK data with only PGCPs did not provide sufficiently accurate data as broad-scale systematic errors were
observed with a pattern sometimes referred to as "bowing" or "doming" which can affect SfM-processed nadir-only imagery (James and Robson, 2014). However, solely using GCPs resulted in slightly reduced vertical accuracy compared to utilizing both GCPs and PGCPs. The PGCPs would have been particularly helpful in remedying potential vertical or horizontal offsets between different models with further georeferencing done using, for example, an iterative closest point (ICP) algorithm, which would provide comparably stable control points regardless of the snow depth. To be practical with the RTK-UAS
workflow, the PGCPs would have to be shaped in a way that discourages the accumulation of snow on top of the PGCP to remove the need for manual cleaning of accumulated snow.

Recently, Revuelto et al. (2021) did a comparison of different UASs in snow depth mapping, including two affordable multicopters (Parrot Anafi and DJI Mavic Pro 2) and a SenseFly eBee Plus fixed-wing, which was also utilized in this study. They concluded that under same illumination conditions, all the tested platforms provide equivalent snow depth products in
terms of accuracy. However, they noted that all the snow depth maps utilized the same snow-free point cloud (acquired by the eBee Plus), and thus are not fully independent. In our case, statistically significant differences were observed between the UASs in each subplot and survey date when utilizing independent UAS-derived snow-free models. When utilizing the non-independent ALS-derived snow-free model, however, significant differences were only observed in the forest subplot or during



lowlight DEC 12 conditions. This clearly highlights the importance of an accurate snow-free model, and how the snow-free
model can be a bottleneck with regards snow depth map accuracy when operating certain UASs (e.g., eBee or Mavic).
Revuelto et al. (2021) also noted that in challenging lighting (overcast sky), all of the UASs failed to properly retrieve the
snow surface. In our case, there were surprisingly no statistically significant differences for any UAS between the lowlight
conditions in December and sunny conditions in February and April when utilizing independent UAS-derived snow-free
models, although a slight increase in accuracy is seen in the DEC 12 to FEB 21 survey. However, we observed that regardless
of the utilized snow-free model, the accuracy was more dependent on land cover type. This seems to be the general trend
observed in other studies utilizing a UAS-SfM approach in snow depth measurements, although most studies report accuracy
only in open areas or do not separate between different vegetation types, with reported RMSEs varying between ~6–58 cm
depending on the study (Vander Jagt et al., 2015; De Michele et al., 2016; Harder et al., 2016; Bühler et al., 2017; Cimoli et
al., 2017; Adams et al., 2018; Avanzi et al., 2018; Revuelto et al., 2021). Buhler et al. (2016) report an RMSE of 7 cm for areas
with short grass and 30 cm for areas with brushes and/or high grass, while Broxton et al. (2020) report an RMSE ~10 cm in
sparsely forested area and ~10–20 cm in densely forested area. Harder et al. (2020) utilized the UAV-SfM approach in mapping
snow depths at study sites classified by vegetation height to open (< 0.5 m), shrub, and tree (> 2 m) covered areas, looking at
two sites each. They report an RMSE of 10–30 cm in open areas, 13–19 cm in shrub areas, and 20–33 cm in tree areas with
dense needleleaf forest having a higher RMSE (33 cm) compared to leaf-off deciduous trees (20 cm).
In our case, none of the GNSS checkpoints or snow stakes were directly under canopy, yet there is a clear difference between
the accuracies obtained for open mire and mixed/forest subplots. It is difficult to determine whether the different studies that
include forested areas have reference datapoints under canopy or between trees. Nevertheless, the general trend seems to be
for forest areas providing less accurate snow depth data. Possible explanations for this may include reduced GNSS accuracy,
understory vegetation, shadows and lighting related issues, and reduced accuracy of SfM procedure for more complex
landscape.
Recent studies have also started to investigate UAS-lidar approach to snow depth mapping, partly due to the greater canopy
penetration of lidar systems compared to SfM. Harder et al. (2020) compared UAV-SfM and UAS-lidar approaches and
reported the UAV-lidar approach as equally successful in penetrating deciduous and needleleaf canopies, although the errors
were larger (RMSE 13–17 cm) in vegetated sites compared to open areas (9–10 cm). The UAS-SfM approach had a wider
variation in errors as discussed above. Jacobs et al. (2021) utilized a more moderately priced UAV-lidar system, about a third
of the price of the system utilized by Harder et al. (2020), and report RMSEs of 1.2 cm for open areas and 10.5 cm for mixed
forest sites consisting of deciduous and coniferous trees. However, the system utilized by Jacobs et al. (2021) had a relatively
short battery life and the total reported survey time of 2 hours for the 9.8 ha survey was relatively high. Dharmadasa et al.
(2022) also utilized a UAS-lidar approach and report RMSEs of 4.3–22 cm for field sites, 7.9–12 cm for deciduous forest sites,
and 19–22 cm for coniferous boreal forest sites. Furthermore, Dharmadasa et al. (2022) argue that remote sensing techniques
alone are not able to provide comprehensive snow depth distribution under a coniferous canopy, despite the increased point
density provided by the UAS-lidar approach when compared to a traditional ALS approach.



## 4.2 Operational challenges and further considerations

UAS platforms and their use in general topographic mapping have been reviewed by Nex and Remendino (2014) and Colomina
and Molina (2014). Typical platform considerations, such as payload, flight speed and wind resistance, are relevant for snow
depth mapping, but there are also some special considerations to be made. First, battery life may drop severely in sub-zero
temperatures. Secondly, as plowed roads may not be available, portability can become an issue depending on whether a
snowmobile or sled is available, or if the crew has to rely on snowshoes or skis. Considering the selection of suitable camera
systems and camera settings for arctic and subarctic conditions, the short days and low light conditions during winters in high
latitudes require extra attention. An emphasis should be placed on selecting a lens with a relatively large aperture and a sensor
which allows a high enough ISO value (i.e., sensor gain) to provide sufficient shutter speeds without compromising the signal-
to-noise ratio. In the case of snow depth mapping in flat light and low contrast situations, the utilization of a near-infrared
camera can further improve SfM image matching (Bühler et al., 2017). Camera system and camera setting considerations
relevant to general UAS-based topographic mapping have been reviewed by Mosbrucker et al. (2017) and O'Connor et al.
445 (2017).

Another key issue is the required horizontal resolution and vertical accuracy. Vertical accuracy become especially important
when the aim is to track relatively small, centimeter to decimeter scale changes in the snow depth. Generally, the accuracy is
mainly affected by camera parameters, such as focal length and sensor resolution (Mosbrucker et al., 2017), and acquisition
parameters, such as flight height and quality of ground control network (Carrivick et al., 2016). Thus, the selection of a suitable
UAS platform and camera for topographic mapping includes multiple tradeoffs when it comes to desired qualities, even more
so in the case of snow depth mapping in harsh, low-light conditions. A large camera sensor with a high resolution should allow
for higher ISO values to be used without compromising the signal-to-noise ratio of gathered imagery and further provides
better accuracy (Mosbrucker et al., 2017). However, a large sensor often increases the camera size and weight, which in turn
affects the battery life, UAS platform size, and the portability of the system. Similarly, a lower flight height increases the
acquired horizontal resolution and vertical accuracy, but also increases the flight time, battery demand, and the number of
images required to cover the area. The higher number of images in turn requires more computational power when generating
the models. Another aspect to consider is local UAS regulations, which might limit flight height and BVLOS (beyond visual
line of sight) flying (Cracknell, 2017; Stöcker et al., 2017).

Different challenges related to the operation of lightweight UASs, such as pre-flight planning, flight operations, weather,
redundancy, data quality and batteries have been comprehensively discussed by, among others, Duffy et al. (2018) and Kramar
et al. (2022). When operating in subarctic winter, weather-related phenomena produce the clearest challenges in terms of
preflight preparations, operation, data quality and battery life. In our case, study site evaluation and preliminary flights were
performed during the spring preceding the study winter to find the optimal flight parameters and suitable take-off and landing
areas. Due to the remoteness of the site, the amount of time and the field crew size required for the ground and aerial surveys
were also considered, with a target of completing each survey within one week. The most notable changes were to the size of



the survey area in order to ensure it could be covered in the available time with the required data quality. This was needed because of the issues with the drone battery life, the short daily aerial survey window due to limited daylight time during midwinter, often-unpredictable weather conditions and deep soft snow conditions slowing down the deployment of GCPs. The planned surveys were postponed several times during the winter due to weather conditions, including snowfall and high wind

speeds that were obviously beyond our control.

During fieldwork, wind and temperature data from local weather stations was used to find optimal time windows for the surveys. However, partly because of the nearby fells, sudden wind gusts forced flight operations to be halted a few times. On one occasion, sudden gusts stopped the autopilot controlled UAS in its place during a mission and manual "tacking" maneuvers were required to bring it back to the takeoff and landing location. The field experience indicated that the more robust drones

(P4A, P4RTK, eBee) could be operated at up to 10 m/s with higher wind/gust speeds, whereas the lighter weight Mavic could not be confidently operated in such conditions.

Furthermore, the topography in the area caused rapid temperature changes, especially in the open mire located in lower elevations between the fells. The survey in January was unsuccessful due to very low temperatures (-30 °C), which caused freezing of the DJI Mavic lens system, leading to unfocused aerial photographs. One potential cause of this may be

condensation and subsequent freezing of water in the lens machinery caused by temperature changes while moving from the warm storage area to a cold car and again to a warm research station premises before flights in cold outside conditions. One possible solution could be to use desiccant bags to reduce moisture and use a more heavily insulated bag for the storage of the UASs to help slow acclimation to a new environment.

UAS manufacturer guidelines usually give general recommendations on the operational temperature range of the UAS

batteries. In cold climates, the lower limit is naturally the concern and significant drops in capacity or even malfunctions can be experienced in low, sub-zero temperatures (Ranquist et al., 2017). Some smart batteries can also have digital warning indicators or even power cutoffs preventing takeoff if the temperature is too low, thus requiring pre-heating. One good option is to store the batteries in a heat box to retain an optimal battery temperature and as much capacity as possible during fieldwork. The UAS powertrain in quadcopters was observed to create enough heat to keep the drone operational in cold temperatures.

However, there might be variability between different models of UAS. Consumer-grade drones are usually certified to operate in above-zero temperatures with some exceptions to above -10 °C (Ranquist et al., 2017). Real-world experience has shown that P4RTKs rated as operating in temperatures between 0 and 40 °C can be fully operated in under -10 °C temperatures.

Temperatures close to 0 °C also caused problems due to moisture, especially for the fixed-wing systems. As highlighted by Revuelto et al. (2021), fixed-wing models that rely on belly-landing, such as eBee, can have issues with rugged or wet surfaces.

We only experienced issues with eBee during the spring melt season, when a malfunction resulted in a loss of data, possibly due to water getting into the electronics during a landing on wet snow. Fixed-wing UASs with VTOL (vertical take-off and landing) capabilities, which have recently become more widely available, may partly mitigate the issues of sub-optimal landing areas, while still retaining the advantages of fixed-wing platform, such as longer flight times and larger mapping area extent compared to multicopters.





There are several advantages to the UAS-lidar approach over the UAS-SfM, including more accurate DEM extraction when flying over homogenous textures and the possibility of better penetration of the tree canopy. Compared to the UAS-SfM approach, which uses using passive RGB camera sensors, UAS-lidar, as an active measurement technique, provides the possibility of night-time operation, which becomes very useful during the winter months in northern latitudes when the day only lasts for a couple of hours. Furthermore, the price of a professional grade UAS-lidar setup is considerably higher than a

professional grade UAS-SfM setup, although prices have dropped noticeably over recent years. Nonetheless, relatively cheap UASs relying on UAS-SfM, such as DJI Mavic Pro, can do the job, especially in more open areas and under good lighting conditions; there is still, however, a need to acquire a suitable RTK-GNSS device to accurately measure the GCPs. A slightly more expensive RTK-equipped UAS can be well worth the extra costs as the need for GCPs are reduced and the data quality is generally superior due to better sensor capabilities and improved georeferencing. There is also an aspect of improved safety

as there is less of a need to place GCPs in difficult terrain and/or conditions and thus fewer work hours are required. Although the UAS-SfM approach cannot detect sub-canopy snow depths, it can provide a much more comprehensive view of the catchment-scale spatial distribution of snow depth compared to manual snow surveys. The findings related to snow-canopy interactions, and variability of snow depth in different landscape units found on the high spatial resolution of the digital snow surface model are comprehensively discussed in the companion paper (Meriö et al. 2022, submitted).

## 5 Conclusions

Our analysis indicates that the measurement accuracy of snow depth using the UAS-SfM approach in subarctic conditions is associated with the i) the UAS platform, ii) land cover type, and, to a lesser degree, iii) light conditions (i.e., flat light vs. direct sunlight) during flights. Significant differences between the UAS platforms were observed with overall RMSEs varying between 13.0 to 25.2 cm, depending on the UAS. However, data from all platforms could be usable in further analysis and to

produce spatially detailed snow depth information, especially during the times in winter when the snow depths relative to the uncertainty of snow depths are high (i.e., high signal-to-noise ratio).

All the tested UAS platforms exhibited increased uncertainty when operated in forest or mixed landscapes compared to open mire areas, even though none of the GNSS checkpoints or snowline measurement points were directly under canopy. A small increase in accuracy was observed when the data collected during low-light polar night conditions was compared to data

collected in brighter conditions in the spring; these differences was generally not statistically significant, however. Of the tested platforms, eBee Plus RTK and DJI Mavic Pro clearly benefitted accuracy-wise from utilizing ALS data for the snow-free model. This highlights the importance of an accurate snow-free model as any large errors will propagate to all the snow depth maps. The DJI Phantom 4 RTK did not see benefits from utilizing ALS data and provided the best data quality in each situation.

Our findings present the potential of the UAS-SfM approach for measuring snow depth spatially and accurately in harsh subarctic conditions and under the influence of canopy structures. We propose that this technology should be further explored



and taken as part of regular snow monitoring schemes to help identify spatial snow cover changes, snow accumulation patterns, and overall snow depths.

## Data availability

The data underlying this analysis and its documentation is available at https://doi.org/10.23729/43d37797-e8cf-4190-80f1-ff567ec62836 (Rauhala et al. 2022) under a Creative Commons CC-BY-4.0 license.

## Author contribution

AR, HM, PA, and LJM designed the field studies, while AR, LJM, AK, and PK carried them out and processed the data. AR analyzed the data and prepared the manuscript with contributions from all co-authors. HM and PA supervised the research.

## Competing interests

The authors declare that they have no conflict of interest.

## Acknowledgements

This study was supported by the Maa- ja vesitekniikan tuki ry, K. H. Renlund Foundation, Academy of Finland (projects 316349, 330319, and ArcI Profi 4), the Strategic Research Council (SRC) decision no. 312636 (IBC-Carbon), EU Horizon 2020 Research and Innovation Programme Grant agreement no. 869471, and Kvantum institute at the University of Oulu. We thank Valtteri Höyky and Metsähallitus for assisting with field sampling campaigns. We gratefully acknowledge the field work assistance of Filip Muhic, Kashif Noor, Aleksi Ritakallio, Alexandre Pepy, Jari-Pekka Nousu and Valtteri Hyöky.

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
