# Peer review of "Figure S1: The difference between the DEM and GNSS survey elevations for each subplot and survey date."

_The Cryosphere, 2022_

## Author Comment (AC2)

**General Comments**

This manuscript presents the retrieval of snow depth using several UAS platform designs with variable optical sensor setups over three landcover types, to test the efficacy of snow depth retrieval in harsh arctic environments. The experiment is exhaustively described and well documented, with very articulate description of the drone setup, the weather during flights, and set up of the ground control points vs RTK positioning. The major result of the paper that is echoed by the community of UAV snow depth retrievals is that with increasing complexity in the canopy, the reduction of accuracy of the retrieved snow depth. However, the authors also present that whether GCPs are included in the scene or RTK positioning correction Is used that the different in error is negligible.

Response: We thank the anonymous referee for the review of the manuscript and the helpful comments and suggestions. Point by point responses (in blue font) are given below.

Overall the paper is very well written and I believe will be ready for publication with minor revision. There are some improvements that can be made with respect to the presentation of results, especially when having the capability to compare individual measurements to their associated snow depth retrieval. In the results section box plots are used to compare UAV-derived snow depths to the DoDs and snow line measurements. There is a place for including the box plot as it does present the distribution of the data, however it is difficult to ascertain where the most disagreement is occurring in the retrievals. I would expect to see a 1:1 scatterplot that compares the UAV-derived snow depths to in-situ observations. That would allow the reader to understand where the highest deviations are occurring. Are they occurring in areas of thin snowpack, or deep snow? I would also recommend showing the deviations on the map – are they locally clustered? Or homogeneously distributed about the map?

Response: Thank you for the suggestion. We can certainly provide the 1:1 scatterplots comparing UAS-derived and in-situ observation snow depths in the supplement. However, the suggested maps are somewhat problematic since it is difficult to display multiple values on one map. Due to the dataset containing three different UAS, two different baselines (snow-free models) for each, three different subplots, and four different dates, the amount of maps would be quite high. We would suggest that plotting the difference between UAS-derived snow depths and in-situ observations ($\Delta hs_t$) against the stake number (or alternatively distance along the transect) would essentially provide the same information. This, while still being able to provide more information on one graph and retaining readability. See examples below.

[Figure]

Fig. Difference between UAS-derived snow depths and in-situ observations (Δhs$_t$) against the stake number with P4RTK.

[Figure]

Fig. Difference between UAS-derived snow depths and in-situ observations (Δhs$_t$) against the stake number with P4RTK-L.

The Discussion section was well written, however quite long. Section 4.2, specifically lines 434 to 470 read more like a review of issues to consider when doing topographic or snow mapping with UAVs, discussion lens focal length, light conditions, horizontal/vertical accuracy, battery life, etc. I would recommend either shortening or removing this section for clarify of the results. However, lines 471 to 499 are relevant to the research being conducted in this manuscript, so I would suggest keeping it.

Response: We will shorten the suggested part and describe the issues more briefly while relying on providing suitable references for further info. Referee #1 asked for more emphasis and focus on recommendations and best practices, such as "the best platform for accuracy/ease of use (RTK vs. GCPs), general recommendations on GCP use, environmental operating suggestions (cold temps and wind), appropriate baselines, and operations in low light conditions". Some of the aspects discussed in lines 434 to 470 are still relevant in that regard.

**Specific Comments:**

Page 2 Line 30: "snowlines" I see this more commonly referred to as snow transects in snow literature. I'm not sure if this is something that requires changing.

Response: In our view, both terms are used, but perhaps this is a more common practice in the Nordic countries. Nevertheless, all instances of "snowlines" changed to "snow transects" as suggested.

Page 2 Line 36: "Homgren" Should be Holmgren.

Response: Misspelling, fixed.

Page 2 Lines 45 – 49: There have also been some UAS work on freshwater lake ice to retrieve snow depth from structure from motion (Gunn et al., 2022), which could also be included in your description of UAS work in arctic conditions on line 54.

Gunn, G. E., Jones, B. M., & Rangel, R. C. (2021). Unpiloted aerial vehicle retrieval of snow depth over freshwater lake ice using structure from motion. Frontiers in Remote Sensing, 2, 675846.

Response: Thank you for the suggestion, citation added.

Page 3 Line 77: "mire" This is the first time you refer to "mire" areas. What are mire areas (aka bogs, wetland).

Response: Mire is commonly used terminology for areas with wet waterlogged soils forming peat. It includes wetlands, peatlands (bogs, fens, aapamires, etc.) and many other types. We kept the terminology 'mire' as it is commonly used and standardised terminology. Sentence now changed to give short description:

"One is an open treeless area with waterlogged peat soils, here referred to as mire area (approx. 14.4 ha)."

Page 6 Line 146: "placed an average of 52 meters apart" – One thing to keep in mind is that the stakes are placed quite far apart – how do you validate the spatial heterogeneity of snow depth when the spatial autocorrelation is typically around 50 meters? Or is that the reason that you're choosing 50m as an average distance apart.

Response: We used a standardised snow line/transect measurement system following the protocol of the Finnish Environmental Institute (Kuusisto, 1984; Lundberg and Koivusalo, 2003). By measuring snow depth in every 50 m, the measurement contains different landcover types and randomly selected measurement points. This reduces autocorrelation in the measurements.

Kuusisto, E.: Snow accumulation and snowmelt in Finland, Publications of the Water Research Institute 55, National Board of Waters, Helsinki, 149, 1984.

Lundberg, A. and Koivusalo, H.: Estimating winter evaporation in boreal forests with operational snow course data, Hydrol Process, 17, 1479–1493, https://doi.org/10.1002/hyp.1179, 2003.

---

## Author Response (AR1)

**Author's response**

Point by point responses to the referee comments are given below in blue font, clarifications to edits made after the replies are given in red font.

**Review 1:**

**General Comments**

The authors present an overview of the methods employed to measure snow depth from UAVs using Structure from Motion (SfM) techniques over a sub-arctic environment. Notably, they present comparisons between several different UAV platforms in terms of both general platform limitations and with respect to measured vs. observed snow depths. Such comparisons are also partitioned by date and land cover across three separate plots.

In my opinion, the paper is well-written and presents results clearly and concisely through the effective use of tables and figures. The methods are well presented to allow repeatability and the various platforms are well described and compared. I think the work's major novelty lies in the inter-comparison of several platforms, and different baselines, as well as a large range of dates, land cover, and illumination conditions. Bringing all of these components together is challenging, but I think it is done well and presented in a way that will allow others to design and implement effective SfM snow depth mapping projects in similar regions.

With that said, I believe there are a few areas that could be improved. This refers to a need for a heightened emphasis on recommendations especially pertaining to the use of multiple baselines. A clear section or figure aggregating the recommendations from throughout the work would greatly benefit the paper. Without the focus on best practices, this work essentially presents a workflow that has already been established to some degree in the literature. More specific suggestions along these lines are presented below.

Response: We thank the anonymous referee for the review of the manuscript and the helpful comments and suggestions. Point by point responses (in blue font) are given below.

**Major Suggestions**

Provide more emphasis on recommendations. This could be in the form of a table or a new figure within the discussion section. Such recommendations that are mentioned in the text include: the best platform for accuracy/ease of use (RTK vs. GCPs), general recommendations on GCP use, environmental operating suggestions (cold temps and wind), appropriate baselines, and operations in low light conditions. Currently, this information is largely spread throughout the text.

Response: We will prepare e.g., a summary table in the discussion section 4.2 for different platforms, GCP strategies, etc., and their respective applicability and suitability denoted for example as (-) poor, (+) suitable (++) good. Further effort will be made to collect recommendations and related information in that section.

The following paragraph with the main recommendations/observations was added to the end of Section 4.2, lines 536–546:

"Finally, some recommendations and observations can be summarized:

- The obtained results indicate that UAS with RTK correction and a single GCP for bias correction can provide sufficient accuracy for snow depth mapping with much less field work involved, thus improving efficiency and safety.
- For considerably larger areas than the subplots (< 20 ha), multiple GCPs would likely be beneficial even with RTK-capable UAS.
- An accurate snow-free model is essential since any errors will propagate to snow depth models.
- Snow-free baseline obtained with ALS can benefit some UAS, especially in complex, forested landscapes.
- Polar twilight can provide enough (directional) light during solar noon and in clear sky conditions for sufficient contrast required in UAS-SfM processing.
- Consumer-grade and professional UAS can be fully operated in under -10 °C temperatures, but care should be taken to keep batteries warm and to avoid quick temperature changes moving outdoors."

The use of different baselines in this work and pros/cons should be highlighted more. I believe more focus should be placed on the tradeoffs between using the ALS and UAV-based baselines as well as recommending best practices. There is novelty and value to adding an increased focus on these particular findings. For example, based on the results, it seems to me that (if possible) establishing a LiDAR-based baseline might be a good standard practice before doing SfM work.

Response: Thank you for the good suggestion. In our case, we were "lucky" that the ALS campaign to update the freely available, national pointcloud/DEM inventory was in the same region during the surveys (updated on average every six years). It is evident that similar freely available, sufficiently high resolution, and up-to-date ALS data is not generally available globally. However, it might indeed be a good practice to acquire LiDAR-baseline, if possible, e.g., by renting a suitable UAS-Lidar equipment to save on the costs. Furthermore, even if relying solely on UAS-SfM, emphasis should be placed on acquiring accurate baseline as the errors will propagate to all snow depth models. During summer in high latitudes, there is a very long window for the flights as the sun never sets, although one should still aim to do the flights during the window when sun is relatively high in the sky. The fieldwork is also in general less demanding during summer. Thus, there should be better opportunities to spend extra time ensuring that the baseline will be captured with high accuracy. For example, bulk of the literature on the accuracies of RTK/PPK-equipped UAS indicates that one GCP could be sufficient to largely mitigate possible bias. However, much of the literature has provided results in areas < 20 ha (similar to subplot sizes in this study). Recently submitted manuscript (Rauhala, 2023) by the first author of this manuscript highlights that accuracy-wise it is be beneficial to use multiple GCPs at least with PPK workflow in larger areas. This is especially relevant with regards fixed wing platforms such as eBee RTK utilized in this study, which are capable of mapping much larger areas compared to smaller multicopter UAS. We will add a paragraph to further discuss the baselines, tradeoffs, and best practices.

Rauhala, A.: Accuracy assessment of UAS photogrammetry with GCP and PPK-assisted georeferencing. Submitted to the Proceedings of FinDrones2023 (New Developments and Environmental Applications of Drones), Springer Remote Sensing/Photogrammetry, 2023.

Following text was included, lines 405–407 in revised manuscript:

"Recent study by Rauhala (2023) highlighted that UAS survey in a 1 km2 area significantly benefitted from utilization of multiple GCPs even when utilizing PPK correction."

Following text was included, lines 424–429 in revised manuscript:

"It might be beneficial to acquire very accurate snow-free model with lidar as a baseline for snow depth mapping, especially in a complex, forested landscape. Another option would be to make extra effort in acquiring very high resolution and high accuracy snow-free model with UAS-SfM utilizing professional UAS, even if the winter measurements would be performed with a more portable platform. Especially in northern locations, there generally are very long daily aerial survey windows as the Sun never sets and the field work is overall less demanding."

Following text was included, lines 470–473 in revised manuscript:

Recent study by Štroner et al. (2023) highlighted that while mapping snow-free forest sites, low-cost UAS-lidar (DJI Zenmuse L1) can produce much better coverage under the canopy, but still has significantly lower vertical accuracy than high-quality UAS-SfM camera (DJI Zenmuse P1) half the price.

The lighting conditions aspect was stated as one of the primary objectives of this study, though I feel it wasn't sufficiently focused on/addressed – It seemed to be shown that low light didn't notably affect the retrievals, but this is in contrast to previous literature. The discussion on this topic seemed to be a bit of an afterthought. Other points may be considered to be emphasized instead (like the choice of baselines or the performance across multiple platforms).

Response: Due to findings in previous literature, we expected stronger effect of the low light conditions on the accuracy of the results. From the image acquisition perspective, when operating the camera at maximum aperture, one can either increase the ISO value or decrease shutter speed to acquire properly exposed images. The former runs the risk of increasing the noise in the images, possibly leading to spurious tie point correlation results in the SfM algorithm. The latter runs the risk of too much motion blur resulting in poor image quality, and consequently, possibly poor model accuracy. In this case, we were able to find a suitable middle ground between the two. The ISO value was increased two stops to ISO400, whereas lower than usual cruise speed was utilized to further allow decreasing the shutter speed to calculated, lowest possible value that should not induce too much motion blur (> 0.5 pix). Another aspect related to low light (or flat light during overcast conditions) mapping of snow is possible lack of features that are necessary for the SfM algorithm to perform sufficiently. This might be affected by the depth of snow, how fresh the snow is, location, etc. For example, thick snow in open alpine location might be quite even and featureless, whereas shallow, early winter snow on forest location may better show features due to spatial variability being low, i.e., the snow surface quite faithfully follows even small-scale forest floor features. We will include further discussion on the lighting conditions.

Following clarifications were added, lines 431–441 in revised manuscript:

"Diffuse lighting during cloudy conditions and homogeneous snow cover, especially immediately after fresh snowfall, results in low contrast that can cause gaps and large outliers in the generated point clouds (Harder et al., 2016; Bühler et al., 2017). Some authors have also noted that direct sunlight and for example patchy snow cover can lead to similar issues due to overexposed pixels, especially if relying on automatically adjusted exposure (Harder et al., 2016). In our case, there were surprisingly no statistically significant differences for any UAS between the lowlight conditions in December and sunny conditions in February and April when utilizing independent UAS-derived snow-free models, although a slight increase in accuracy is seen in the DEC 12 to FEB 21 survey. This could be explained by the polar twilight (civil twilight) providing enough directional light to create sufficient contrast during solar noon and clear sky conditions. Also, the low snow depths with less spatial variability during the early winter results in a feature-rich snow surface due to the natural variability in the forest floor topography and vegetation. Further steps were taken to adjust the flight speed and camera parameters to allow lower shutter speed and only a slight increase in ISO value (max. two stops to ISO400) to keep the image noise tolerable."

**Minor/Technical Suggestions**

L15: revise - "...and the UAV snow depth results compared to in situ measurements."

**Response: Revised as suggested.**

L24 - L31: Try to separate the sentences on societal impacts & ecological/environmental implications. Reads as though it could be plant/animal communities OR human communities. Also, generally, can add a bit more detail here on the importance of snow before moving on to monitoring approaches. Especially, how snow relates to nature/the environment.

**Response: Done, introduction was modified as suggested:**

"Knowledge of changes in snow accumulation, depth and melt is crucial for nature and society in northern and alpine regions. In the northern hemisphere especially, snow is important to local ecology providing shelter and protection from harsh winter conditions and supporting early summer hydrological conditions (soil moisture, discharge, etc) and a unique environment in north and mountainous areas (Demiroglu et al., 2019; Boelman et al., 2019). Also Northern communities, tourism and industry are adapted and often depended on snow conditions as winter resources (transport, leisure) but also as water storage for hydropower and other needs. Currently, snow resources are threatened by global warming, which will have many direct and indirect effects on northern environments (Carey et al., 2010, Bring et al., 2016). Any changes in magnitude, timing and variability of snowfall, accumulation patterns and melting will alter, among other things, water availability and soil moisture (Barnett et al., 2005; Kellomäki et al., 2010), which, in turn, impacts flood prediction and warning, hydropower generation (reservoir inflow forecasting), water management, transportation, local authority daily management activities and the tourism sector (Veijalainen et al., 2010)."

L45 - L52: Either provide a bit more of an overview on the range of what was done in these studies or point the reader to the latter discussion section where you detail these studies more.

Response: Changed the introduction of previous studies as follows:

"Numerous studies have assessed the potential of using UASs in snow depth mapping during recent years in various locations including alpine (Vander Jagt et al., 2015; Bühler et al., 2016; De Michele et al., 2016; Bühler et al., 2017; Adams et al., 2018; Avanzi et al., 2018; Fernandes et al., 2018; Redpath et al., 2018; Revuelto et al., 2021), apline and prairie (Harder et al., 2016; Harder et al., 2020), meadow and forest (Lendzioch et al. 2016; Broxton and van Leeuwen, 2020), arctic (Cimoli et al., 2017), and freshwater lake settings (Gunn et al., 2021)."

L59: 'submitted to the same journal' can be removed

Response: Removed, as suggested.

L59 - L60: consider rewording, the data itself shouldn't have implications; are you referring to the insights it provides, regarding accumulation and melt patterns?

Response: We are referring to insights, sentence changed as following:

"The accompanying paper (Meriö et al. 2022) delves deeper into the insights provided by the gathered data on local snow accumulation and melting patterns."

L159: The tree masking is an interesting approach. I suggest adding an additional sentence or two describing the 'Maximum Likelihood Supervised Classification'. Additionally, assuming there is a related citation/paper to this approach, that should be included here.

Response: Included a citation and changed as follows:

"The three masks were generated using Maximum Likelihood Supervised Classification, which is a probabilistic approach derived from the Bayes theorem (Ahmad and Quegan, 2012). In the classification, each pixel is assigned to one of the desired classes it has the highest likelihood of belonging, based on the training samples."

Ahmad, A. and Quegan, S.: Analysis of maximum likelihood classification on multispectral data, Applied Mathematical Sciences, 6, 6425–6436, 2012.

Figure 2 (and throughout): The use of the term DEMs of Difference (DoD) is used throughout. I suggest adding the acronym in the figure

Response: Acronym will be added to Figure 2.

L174: Related to the previous point, make sure that the difference (or not) in meaning between DoD and snow depth is clear (and why DoD is used instead of snow depth). Are these terms interchangeable?

Response: DEM of Difference (DoD) is a commonly used term, especially in geosciences/geomorphometry literature, describing the result of differencing between two digital elevation models (or digital surface models). In this context, UAS-derived snow depth map and DoD can be used interchangeably. The advantage of using the term DoD is that it simultanously describes at least part of the method by which the snow depth map was

acquired, while also highlighting that the accuracy of the map is dependent on both the accuracy of the baseline and the accuracy of the obtained snow surface elevation (i.e., the concept of error propagation). We included a clarification that in this study DoD is used interchangeably with UAS-derived snow depth map.

Sentences were changed as follows, lines 183–186 in revised manuscript:

"After manual cleanup of a few classification errors, the masks were utilized for canopy removal before subtracting the snow-free (bare ground) DSM from each snow-covered DSM, thus producing the the DEMs of difference (DoD) highlighting the snow depth. Finally, the DoDs (used interchangeably with snow depth map) were aggregated to 50 cm per pixel resolution before further analysis."

L215: Provide citation/source for "Levene's test"

Response: Citation added (Levene, 1960).

Levene, H.: Robust tests for equality of variances, Contributions to Probability and Statistics: Essays in Honor of Harold Hotelling, 69, 1960.

L225: 'struggles' -> 'performs poorly' (or similar)

Response: Changed to 'performs poorly' as suggested.

L240: are these biases high or low? It may also be valuable to add a table with information of the bias/errors of all the baselines used (maybe in supplementary material)

Response: These biases are high relative to e.g., baselines obtained with P4RTK, which had biases of -0.64, 1.82, and 3.66 cm for mire, mixed, and forest subplots, respectively (see table below). We can include statistics for all baselines (or even all surveys) in the supplement.

|                   | P4RTK | Mavic | eBee  | ALS  |
|-------------------|-------|-------|-------|------|
| Bareground mire   | -0.64 | 7.82  | 16.56 | 0.64 |
| Bareground mixed  | 1.82  | 5.37  | 9.97  | 0.83 |
| Bareground forest | 3.66  | 2.67  | 4.34  | 7.46 |

Table. Biases/mean errors for baselines obtained with different platforms.

This is also a good place to point out that Eq. (3) and the resulting Table 4 contained unfortunate errors. Since the DoDs are obtained by subtracting the baseline models from the snow surface DSMs, it is in our view more reasonable to subtract the mean error of baseline from the mean error of snow DSM to obtain a trueness estimation for individual DoDs (i.e., snow depth maps). Especially so as the DSMs have a general tendency to have a positive bias (40 times out of 45 when the four different baselines are included). This way the trueness estimations highlight that the positive biases to a degree cancel each other out and e.g., a negative DoD trueness indicates likely underestimation of snow depth, which is often seen in the comparison of UAS-based and manual snow depth measurements in Fig. 8 boxplots. Revised Table 4 is given below.

Table 4. DoD trueness during different measurement periods and in study locations.

|                   | MIRE  |       | MIXED  |       |       | FOREST |       |       |       |
|-------------------|-------|-------|--------|-------|-------|--------|-------|-------|-------|
|                   | P4RTK | Mavic | eBee   | P4RTK | Mavic | eBee   | P4RTK | Mavic | eBee  |
| DEC12 (cm)        | 4.13  | -3.73 | -13.18 | 3.61  | -2.64 | -7.70  | -0.06 | -1.56 | -4.12 |
| FEB21 (cm) | -0.04 | -5.57 | -16.52 | -0.37 | -5.71 | -8.24  | -0.14 | 9.62  | 3.09  |
| APR03 (cm)        | 4.14  | -3.28 | -15.78 | -1.53 | -4.26 | -6.80  | -4.76 | -0.89 | -9.60 |
| APR24 (cm)        | 0.58  | -7.48 |        | -0.60 | -1.27 |        | -6.03 | 3.86  |       |

Mean errors, standard deviations, and root mean square errors for all generated digital surface models were included in the Supplement, Tables S1–S3.

Figure 1: If you are going to present the point snow depth observation from the ultrasonic sensor (in Figure 7), make sure to indicate its location in Figure 1

Response: Location will be added to Figure 1.

L294: 'No clear correlation' -> can you describe this a bit more? It is concerning if the observed and UAV-derived depths are not correlated across all depth products. That draws into question the ability of these approaches to accurately capture the snow variability across the basin, and only suggests it can capture general differences through time. Please add some more interpretation here.

Response: This was a good remark, the description of correlations was indeed badly worded or rather described too briefly. Strong and statistically significant correlations were not observed on all individual dates or subplots. This is especially true for the DEC12 survey. This is likely due to the snow depth being quite uniform in the early winter. As described in the manuscript, the "manual snowline measurements resulted in mean snow depths and standard deviations of 36.8 cm and 4.8 cm for the DEC 12 survey, …". Thus, the small variations in the snow depth are not captured well due to inherent random errors in the UAS-based snow depth measurements. This was also reflected on in the conclusions: "However, data from all platforms could be usable in further analysis and to produce spatially detailed snow depth information, especially during the times in winter when the snow depths relative to the uncertainty of snow depths are high (i.e., high signal-to-noise ratio)." If we for example look at the correlation coefficients for P4RTK during different surveys from DEC12 to APR24 (Table below), we see clear increase in correlation as the winter progresses, snow depths increase, and/or local variations get larger. With the flooding mire subplot excluded, the correlation coefficient for P4RTK during APR24 would be as high as 0.86.

|       | P4RTK | P4RTK-L | eBee  | eBee-L | Mavic | Mavic-L |
|-------|-------|---------|-------|--------|-------|---------|
| DEC12 | 0.05  | 0.18    | -0.16 | 0.00   | -0.20 | -0.12   |
| FEB22 | 0.39* | 0.47*   | 0.31  | 0.52*  | 0.12  | 0.18    |
| APR03 | 0.68* | 0.59*   | 0.49* | 0.60*  | 0.58* | 0.70*   |
| APR24 | 0.68* | 0.59*   |       |        | 0.54* | 0.61*   |

Table X: Pearson correlation coefficients for different dates.

\*Statistically significant at the p<0.05 level

We will include the table and following clarification is given:

"In the early snow season with low snow depths, no clear correlation was observable on individual subplots/dates when comparing manual snow depth measurements and the UAS-derived snow depth pixels at the location of the manual measurement (Table X). This is likely due to the snow depth being quite uniform in the early winter. Thus, the small variations in the snow depth are not captured well due to inherent random errors in the UAS-based snow depth measurements."

The above Table is included (Table 7) and following clarification is given, lines 305–310:

"In the early snow season with low snow depths, no clear correlation was observable on individual subplots/dates when comparing manual snow depth measurements and the UAS-derived snow depth pixels at the location of the manual measurement (Table 7). This is likely due to the snow depth being quite uniform in the early winter. Thus, the small variations in the snow depth are not captured well due to inherent random errors in the UAS-based snow depth measurements. A clear increase in correlation is observed when the winter progresses, and the snow depths increase and/or local snow depth variability increases. If the flooding mire subplot during APR24 is excluded, the correlation coefficient for P4RTK would be as high as 0.86."

Figure 9: In the results, can you add a comment on the clear high (Dec 12) and low biases (Apr 3)?

Response: We added the following comment regarding the mentioned biases:

"There is noticeable positive bias with all UAS during the DEC 12 survey and a negative bias during the APR 03 survey. To a degree, these biases are also seen in the comparison of generated DSMs and checkpoints, especially in the case of DEC 12 survey where all the UASs had a tendency for overestimating the snow surface elevation."

L401 – 404: This is a good opportunity to delve into why you think the lowlight conditions did not affect the products notably (and why it did in Revuelto et al 2021). The stated objective of providing an assessment of how low light conditions affect SfM snow depth mapping is left with a somewhat ambiguous conclusion. Make sure this is clear.

Response: We addressed this earlier, discussion on the lowlight conditions will be included.

L470: Reword – I think a general statement mentioning how weather constraints and unpredictability are unavoidable limitations when working with UAVs + providing your example.

Response: Reworded as follows:

"Unpredictable weather places unavoidable limitations on all UAS operations. In our case, the planned surveys were postponed several times during the winter due to weather conditions, including snowfall, very low temperatures, and high wind speeds."

Conclusion: One recommendation - specifically recommend a platform (i..e, the Phantom 4 RTK) for providing the best SfM products (w/ relevant error statistics)

Response: We will make a platform/workflow recommendation to use Phantom 4 RTK or similar platform with at least one GCP on areas comparable to subplot sizes in this study. For considerably larger areas the recommendation is to use multiple GCPs if possible (following the best practices addressed earlier).

Following recommendation was added, lines 562–564 in revised manuscript:

"The DJI Phantom 4 RTK provided the best overall accuracy (RMSE 13.0 cm) and correlation (r = .89) with the manual snow course measurements. Recommendation can be made for similar platform utilizing RTK or PPK with at least one GCP for snow depth mapping in areas of similar size (< 20 ha)."

**Review 2:**

**General Comments**

This manuscript presents the retrieval of snow depth using several UAS platform designs with variable optical sensor setups over three landcover types, to test the efficacy of snow depth retrieval in harsh arctic environments. The experiment is exhaustively described and well documented, with very articulate description of the drone setup, the weather during flights, and set up of the ground control points vs RTK positioning. The major result of the paper that is echoed by the community of UAV snow depth retrievals is that with increasing complexity in the canopy, the reduction of accuracy of the retrieved snow depth. However, the authors also present that whether GCPs are included in the scene or RTK positioning correction Is used that the different in error is negligible.

Response: We thank the anonymous referee for the review of the manuscript and the helpful comments and suggestions. Point by point responses (in blue font) are given below.

Overall the paper is very well written and I believe will be ready for publication with minor revision. There are some improvements that can be made with respect to the presentation of results, especially when having the capability to compare individual measurements to their associated snow depth retrieval. In the results section box plots are used to compare UAV-derived snow depths to the DoDs and snow line measurements. There is a place for including the box plot as it does present the distribution of the data, however it is difficult to ascertain where the most disagreement is occurring in the retrievals. I would expect to see a 1:1 scatterplot that compares the UAV-derived snow depths to in-situ observations. That would allow the reader to understand where the highest deviations are occurring. Are they occurring in areas of thin snowpack, or deep snow? I would also recommend showing the deviations on the map – are they locally clustered? Or homogeneously distributed about the map?

Response: Thank you for the suggestion. We can certainly provide the 1:1 scatterplots comparing UAS-derived and in-situ observation snow depths in the supplement. However, the suggested maps are somewhat problematic since it is difficult to display multiple values on one map. Due to the dataset containing three different UAS, two different baselines (snow-free models) for each, three different subplots, and four different dates, the amount of maps would be quite high. We would suggest that plotting the difference between UAS-derived snow depths and in-situ observations ( $\Delta$ hst) against the stake number (or alternatively distance along the transect) would essentially provide the same information. This, while still

being able to provide more information on one graph and retaining readability. See examples below.

Fig. Difference between UAS-derived snow depths and in-situ observations ( $\Delta hs_t$ ) against the stake number with P4RTK.